# AgentRecBench: Benchmarking LLM Agent-based Personalized Recommender Systems

**Yu Shang**[1]* **Peijie Liu**[1]* **Yuwei Yan**[2] **Zijing Wu**[3] **Leheng Sheng**[4] **Yuanqing Yu**[1]
**Chumeng Jiang**[1] **An Zhang**[3] **Fengli Xu**[1]† **Yu Wang**[1] **Min Zhang**[1] **Yong Li**[1]†

[1]Tsinghua University
[2]The Hong Kong University of Science and Technology (Guangzhou)
[3]University of Science and Technology of China
[4]National University of Singapore

## Abstract

The emergence of agentic recommender systems powered by Large Language Models (LLMs) represents a paradigm shift in personalized recommendations, leveraging LLMs' advanced reasoning and role-playing capabilities to enable autonomous, adaptive decision-making. Unlike traditional recommendation approaches, agentic recommender systems can dynamically gather and interpret user-item interactions from complex environments, generating robust recommendation strategies that generalize across diverse scenarios. However, the field currently lacks standardized evaluation protocols to systematically assess these methods. To address this critical gap, we propose: (1) an interactive textual recommendation simulator incorporating rich user and item metadata and three typical evaluation scenarios (classic, evolving-interest, and cold-start recommendation tasks); (2) a unified modular framework for developing agentic recommender systems; and (3) the first comprehensive benchmark comparing over 10 classical and agentic recommendation methods. Our findings demonstrate the superiority of agentic systems and establish actionable design guidelines for their core components. The benchmark environment has been rigorously validated through an open challenge and remains publicly available with a maintained leaderboard at `https://tsinghua-fib-lab.github.io/AgentSocietyChallenge/pages/overview.html`. The benchmark is available at: `https://huggingface.co/datasets/SGJQovo/AgentRecBench`.

## 1 Introduction

Recommender systems have become indispensable for addressing information overload on web platforms by modeling user preferences. Over the decades, the field has evolved from rule-based recommendation [1, 2] to data-driven deep learning approaches [3–5], achieving remarkable gains in accuracy and efficiency. Despite these advances, there still remains some critical challenges, including (1) the black-box nature of recommendations limits interpretability of recommendation results [6, 7]; (2) most methods rely predominantly on historical interaction data, failing to fully leverage rich contextual information in real-world applications [8]; and (3) existing methods often depend on handcrafted features with fixed recommendation strategies [9], which constrain their adaptability to different scenarios.

Recent advancements in LLM-based agents present a transformative opportunity to revolutionize the paradigm of recommender systems [10–14]. Equipped with complex reasoning and role-playing

---

*Equal contribution.

†Corresponding author, correspondence to fenglixu@tsinghua.edu.cn, liyong07@tsinghua.edu.cn.

39th Conference on Neural Information Processing Systems (NeurIPS 2025) Track on Datasets and Benchmarks.

capabilities brought by LLMs, LLM-based agents can automatically tackle complex tasks through human-like planning [15, 16], reflective reasoning [17, 18], and iterative action [19, 20]. Compared to traditional recommender systems, agentic recommender systems offer several key advantages. Firstly, by leveraging natural language as a medium, these systems can explicitly articulate their recommendation logic, enhancing transparency and fostering user trust. Secondly, with advanced perception and reasoning abilities, agentic recommender systems can autonomously gather and integrate rich, personalized contextual information from the environment, beyond ID-based interactions to enrich user modeling. Thirdly, these systems exhibit continuous self-improvement through environmental interactions, dynamically refining their recommendation strategies by incorporating external feedback and internal memory to achieve unprecedented adaptability across various scenarios.

Despite this promising paradigm shift, the field currently lacks standardized benchmarks to systematically evaluate agentic recommendation approaches, which hinders our understanding of their effectiveness and practical deployment. Existing recommendation benchmarks are typically static and non-interactive, rendering them inadequate for modern recommendation agents that require autonomous interaction and dynamic information gathering. To bridge this critical gap, we introduce AgentRecBench, the first comprehensive benchmark for agentic recommender systems, explicitly designed to overcome the limitations of static evaluation frameworks. Our benchmark makes four key contributions to advancing agentic recommender systems. First, to address the central challenge of building an interactive environment, we construct a unified textual interaction environment by processing Yelp*, GoodReads†, and Amazon‡ datasets into a standardized schema. Crucially, we developed standardized information retrieval tools that empower agents to perform flexible and autonomous information retrieval, such as navigating user, review, and item networks to dynamically fetch contextual semantic information for personalized recommendations. This capability is a fundamental differentiator from prior static benchmarks. Second, to tackle the challenge of creating diverse and complex evaluation demands, we establish three carefully designed evaluation scenarios: classic recommendation tasks for general performance assessment, evolving-interest tasks to evaluate dynamic adaptation capabilities, and cold-start tasks to measure generalization ability. These scenarios collectively reflect the core challenges in real-world recommendation systems. Third, we propose a modular agent framework with essential cognitive components [21, 22] for rapidly building agentic recommender systems. The framework includes four core modules: dynamic planning for task decomposition, complex reasoning for decision-making, tool utilization for environment interaction, and memory management for experience retention and utilization. Finally, through extensive evaluation of over 10 existing recommendation agents and traditional methods, we establish the first comprehensive benchmark for agentic recommenders. Our analysis reveals critical insights into current capabilities and limitations, from which we derive practical design guidelines to facilitate future development in this emerging field.

The benchmark's practical utility has been demonstrated through the AgentSociety Challenge [10], which attracted 295 competing teams worldwide and received over 1,400 submissions during the 37-day competition. Participants achieved 20.3% performance improvement in Recommendation Track during the Development Phase, with a further 15.9% gain in the Final Phase, validating the practical utility of our evaluation framework. The complete benchmark environment remains publicly available with continuously maintained leaderboards to support ongoing research.

In summary, this work has the following main contributions:

- We propose AgentRecBench, the first large-scale and comprehensive benchmark that systematically evaluates both emerging agentic recommender systems and traditional recommendation methods across diverse scenarios.

- We provide a textual environment simulator equipped with multi-domain recommendation datasets and a standardized agent development framework, establishing a closed-loop development-evaluation pipeline. This toolkit helps facilitate rapid prototyping and systematic testing of recommendation agents.

- Through in-depth analysis of over 10 approaches, we distill key insights into the superior designs of current approaches. Our findings provide actionable design guidelines to inspire more powerful agentic recommender systems.

---

*https://www.yelp.com/dataset
†https://sites.google.com/eng.ucsd.edu/ucsdbookgraph/home
‡https://amazon-reviews-2023.github.io/

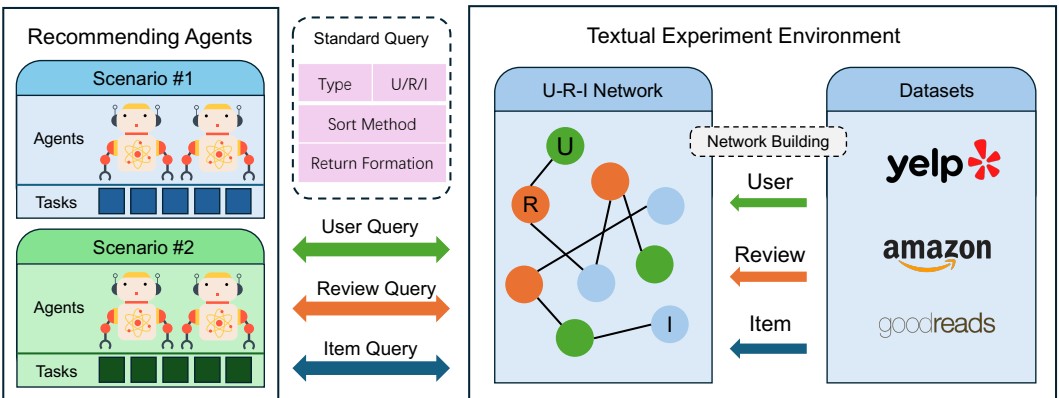

Figure 1: The overall framework of our interactive textual environment simulator.

## 2  Definition of Agentic Recommender System

We formalize an *agentic recommender system* as a system where an LLM-based agent interacts with an environment $\mathcal{E} = (\mathcal{U}, \mathcal{I}, \mathcal{H})$ to produce recommendations. Here, $\mathcal{U}$ and $\mathcal{I}$ denote the user and item feature spaces respectively, while $\mathcal{H} = \{(u, i, c, \tau)\}$ represents user-item historical interactions with contextual metadata $c$ (e.g., ratings, comments) and timestamp $\tau$. At each step $t$, the agent observes a state $s_t \in \mathcal{S}$ (encoding user and item profile, interactions, and other environmental feedback) and selects an action $a_t \in \mathcal{A}$ via policy $\pi_\theta$:

$$a_t \sim \pi_\theta(a|s_t), \quad \pi_\theta : \mathcal{S} \to p(\mathcal{A}), \tag{1}$$

where $p(\mathcal{A})$ is a probability distribution over the action space $\mathcal{A} = \mathcal{I} \cup \mathcal{A}_{seek}$ including item recommendations and information-seeking actions, and $\theta$ parameterizes the agent's internal functional modules such as reasoning and memory.

The agentic recommender system exhibits three distinguishing characteristics that differentiate it from traditional approaches. First, it can adaptively collect personalized contextual information from the environment while maintaining generalized, zero-shot adaptable recommendation policies across diverse scenarios. Second, the system demonstrates self-improving capability by continuously refining its internal representations and decision strategies through accumulated experience and external feedback, enabling ongoing performance enhancement. Third, the system is promising to enhance proactive engagement by strategically initiating targeted interactions beyond passive response paradigms.

## 3  Textual Environment Simulator

The textual experiment environment simulates the information retrieval logic and functionalities typically found in social and web platforms. It provides a standardized interaction space, enabling recommendation agents to query and receive feedback systematically. This environment addresses two primary concerns: defining a clear interaction space and ensuring controlled data accessibility. The overall framework is shown in Figure 1.

### 3.1  Data and Interaction Space

A clearly defined interaction space is crucial for the reliable evaluation of recommending agents. It enables consistent comparisons across various recommendation methods by providing uniform conditions. To facilitate this, we merge diverse data sources into a coherent network structure, forming a unified User-Review-Item (U-R-I) network with standardized query capabilities.

Building the U-R-I network involves aggregating data from multiple sources, such as user profiles, item characteristics, and user-generated content (e.g., reviews and ratings). User nodes represent individuals interacting with the platform, item nodes represent recommendable entities, and review nodes encapsulate user feedback and ratings. Edges between these nodes illustrate interactions, forming a structured and navigable data space for agent exploration.

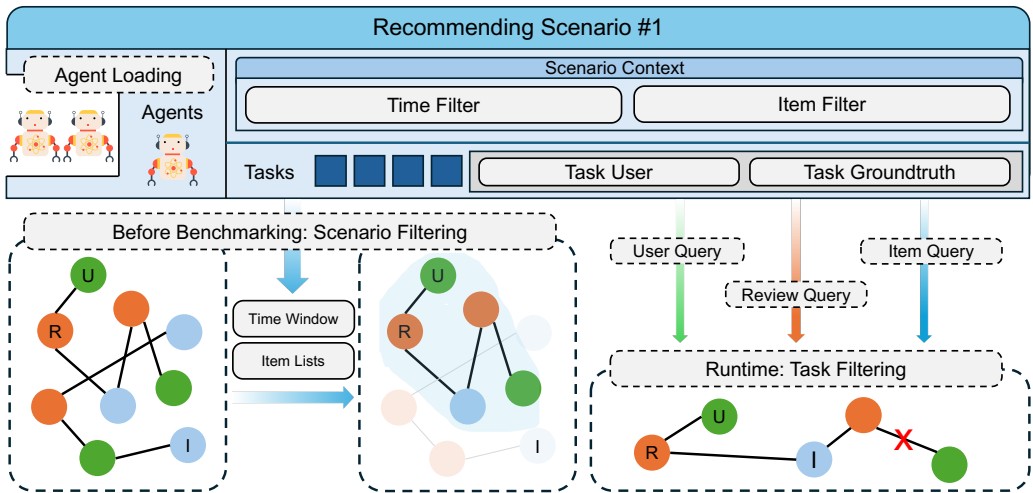

Figure 2: Illustration of the dynamic data visibility control workflow of our textual environment simulator.

The standardized query functionality is defined as:

$$Query(Type, SortMethod, Formation) \rightarrow StructuredData \,|\, TextualData \qquad (2)$$

In this function, $Type$ specifies the retrieval data type (e.g., user, item, review). $SortMethod$ defines how the query results are ordered, such as by date, relevance, or popularity. The boolean $Formation$ indicates the format of the returned data (structured or textual); structured queries yield clearly defined attributes suitable for algorithmic processing(key-values), while textual queries return natural-language content for interpretive tasks.

However, different recommendation scenarios may necessitate various data control mechanisms, such as temporal filtering (e.g., restricting data to a specific time frame), item-based filtering (e.g., selecting items with minimum review thresholds), or user attribute-based constraints.

## 3.2 Dynamic Data Visibility Control

Dynamic data visibility control aims to address the scenario-specific data requirements outlined above. As shown in Figure 2, the core concept is a two-layer control system composed of scenarios and tasks. Scenarios define broad filtering criteria, shaping the environment for a collection of related tasks, while each task focuses on a clearly defined recommendation objective with explicit targets and contexts.

A scenario is formally structured as follows:

$$Scenario = \{TimeFilter, ItemFilter\}, \qquad (3)$$

where $TimeFilter$ manages the temporal scope of accessible data, such as selecting reviews posted within a designated period. $ItemFilter$ sets criteria for item inclusion, like enforcing a minimum review threshold to ensure sufficient data quality. Within each scenario, a task specifies a precise recommendation problem:

$$Task = \{TargetUser, GroundTruth\}, \qquad (4)$$

where $TargetUser$ identifies the individual for whom recommendations are generated, and $GroundTruth$ provides benchmark data (e.g., known user preferences or past interactions) essential for evaluating recommendation accuracy. The proposed dynamic data visibility control ensures flexible, scenario-driven management of data accessibility, facilitating comprehensive and systematic evaluation of agentic recommender systems across diverse experimental conditions.

Table 1: Structure of aggregated data including user-side, item-side profiles and review information.

| Data Type | Description and Key Fields |
|---|---|
| User | • User ID
• Review count
• Social connections (friendship)
• Average rating |
| Item | • Item ID
• Item name
• Type (product/business/book)
• Metadata (price, title, descriptions, location etc.)
• Average rating
• Review count |
| Review | • Review ID
• User ID
• Item ID
• Rating (1-5)
• Textual review content
• Timestamp of interaction
• Helpfulness votes (funny/useful/cool) |

# 4 Experimental Setting

Our experimental evaluation leverages three large-scale publicly available datasets (Yelp[§], GoodReads[¶], and Amazon[‖]), which collectively provide extensive user interaction records and rich user/item profiles that enable comprehensive retrieval and personalization analysis. Below, we provide a detailed description of datasets, defined task scenarios, baselines, and evaluation metrics.

## 4.1 Dataset

The three used datasets are introduced as follows:

**Amazon.** The Amazon dataset captures user purchase behavior, product reviews, and ratings across multiple categories on an e-commerce platform. It comprises large-scale user-item interaction records along with detailed textual reviews, providing a rich source for modeling consumer preferences.

**GoodReads.** The Goodreads dataset consists of user ratings and reviews of books, reflecting diverse reading interests and preferences. This dataset is particularly valuable for studying recommendation models in the context of literary content.

**Yelp.** The Yelp dataset contains extensive user reviews and ratings of local businesses, including restaurants and retail stores. It effectively captures real-world consumer experiences and preferences across a variety of service domains.

We organize these datasets into three structured sub-datasets (items, users, and reviews) to facilitate effective agent interaction and enhance the extraction of user and item features. This structured partitioning enables efficient information retrieval for personalized recommendations. Table 1 summarizes the key fields of the structured dataset. As evidenced in Figure 3 (a), the dataset exhibits complementary domain coverage, with Yelp contributing the majority of users and Goodreads providing the most extensive items. The power-law distributions in user/item interactions shown in Figure 3 (b) and (c) faithfully reflect real-world recommendation scenarios where most users engage moderately while only a few exhibit extremely high activity levels.

---

[§]https://www.yelp.com/dataset
[¶]https://sites.google.com/eng.ucsd.edu/ucsdbookgraph/home
[‖]https://amazon-reviews-2023.github.io/

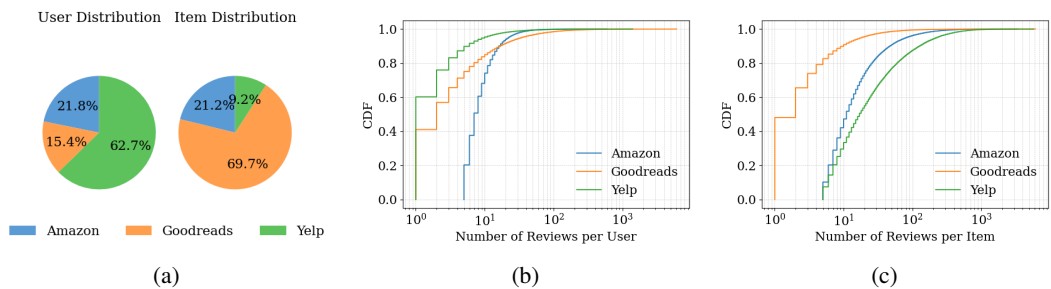

Figure 3: Statistical distributions of our aggregated multi-platform dataset showing (a) user/item distribution (b) user interaction distribution (c) item interaction distribution.

## 4.2 Task Scenarios

To systematically evaluate agentic recommender systems, we design three representative evaluation scenarios that mirror real-world recommendation challenges. We provide detailed descriptions of each scenario below.

**Classic recommendation.** This serves as our foundational benchmark, evaluating general performance under standard conditions where agents can access complete user profiles and interactions from the environment.

**Evolving-interest recommendation.** To rigorously evaluate the temporal adaptation capabilities of agentic recommender systems, we design a multi-scale evaluation framework that captures both gradual and immediate preference dynamics. The **long-term recommendation** task uses a three-month user interaction window. We sample active users ($\geq$5 interactions), testing the system's ability to identify stable user preference. This task allows assessment of the agent's capacity for longitudinal user modeling. Complementing this, the **short-term recommendation** task focuses on immediate adaptation through a compressed one-week interaction window. This task mainly measures agents' ability to model emerging interaction patterns.

**Cold-start recommendation.** To systematically assess the capability of agentic recommender systems in addressing real-world data sparsity challenges, we design comprehensive evaluations for both user-side and item-side cold-start scenarios. In the **user cold-start recommendation**, we construct test sets comprising users with fewer than m historical interactions (where m is dataset-dependent). This setting rigorously examines the system's ability to generate accurate recommendations from sparse interaction signals and effectively utilize available contextual user data. For the **item cold-start recommendation**, we evaluate on items with fewer than n recorded interactions, testing the system's proficiency in reasoning about item attributes and detecting similarity patterns with existing items. This evaluation specifically measures how agentic systems transcend the limitations of conventional ID-based recommendation approaches through item information mining capabilities.

## 4.3 Baselines

We conduct a comprehensive evaluation across three paradigms of recommender systems:

**Traditional Recommender Systems.** We select Matrix Factorization (MF) [1] as a representative traditional recommendation method, which decomposes the user-item interaction matrix into low-dimensional latent factors to capture implicit user preferences and item characteristics.

**Deep Learning-based Recommender Systems.** LightGCN [4] represents an effective modern graph learning-based recommendation approach, employing simplified graph convolution operations to effectively propagate and aggregate neighborhood information in user-item interaction graphs. BC Loss [23] is a bias-aware contrastive loss for collaborative filtering that introduces adaptive margins to mitigate popularity bias and improve recommendation quality. XSimGCL [24] is a deep learning–based recommendation method that uses graph contrastive learning with noise-based augmentation for self-supervised representation learning.

**Agentic Recommender Systems.** Our evaluation includes eight different available agentic recommender systems, which are detailed as follows:

- **BaseAgent**. A handcrafted base agent that performs direct recommendation generation through standard LLM inference without additional reasoning components.

- **CoTAgent**. Extends BaseAgent with zero-shot chain-of-thought (CoT) [25] prompting to enable explicit recommendation rationale generation and step-by-step decision making.

- **MemoryAgent**. Augments BaseAgent with a memory mechanism [16] that maintains and retrieves relevant interaction history to inform recommendations.

- **CoTMemAgent**. Combines the reasoning capabilities of CoTAgent with the contextual retention of MemoryAgent through integrated chain-of-thought reasoning with memory augmentation.

- **Baseline666**. The winning solution from the AgentSociety Challenge Recommendation Track [10], distinguished by its platform-aware feature extraction that dynamically adapts feature representation based on data source characteristics.

- **RecHackers**. The second-place solution from the AgentSociety Challenge Recommendation Track [10], which achieves robust content-based recommendations through the comprehensive integration of user historical comments with detailed item attributes.

- **DummyAgent**. The third-place solution from the AgentSociety Challenge Recommendation Track [10], which employs advanced comment feature engineering to identify and leverage high-information content for enhanced user/item modeling.

- **Agent4Rec** [12]. An advanced framework that implements multi-step reasoning through orchestrated LLM-agent interactions for complex recommendation scenarios.

### 4.4 Envaluation Metric

We evaluate recommendation performance using ranking-based metrics with emphasis on Top-$N$ accuracy. Following standard evaluation protocols [1, 3], each test instance consists of 20 candidate items: one ground-truth positive item sampled from the user's interaction history and 19 negative items sampled from unobserved interactions. The primary metric is *Hit Rate@$N$* (HR@$N$), measuring the probability that the ground-truth item appears in the top-$N$ ranked positions ($N \in \{1,3,5\}$). Formally:

$$\text{HR@}N = \frac{1}{|\mathcal{T}|} \sum_{t \in \mathcal{T}} \mathbb{I}(p_t \in \mathcal{R}_t^N), \tag{5}$$

where $\mathcal{T}$ is the test set, $p_t$ is the ground-truth positive item for test case $t$, $\mathcal{R}_t^N$ denotes the top-$N$ recommendations, $\mathbb{I}(\cdot)$ is the indicator function.

## 5 Experimental Results

In this section, we present the performance of several representative baseline methods evaluated on the AgentRecBench benchmark. Our analysis is structured into three key parts: (1) Overall Performance, (2) Performance in Cold-start Scenarios, and (3) Performance under Evolving User Interests.

### 5.1 Main Performance

We begin by evaluating model performance in the classic recommendation scenario. Experiments are conducted on three datasets using three proprietary model families: Qwen-72 B-Instruct, DeepSeek-v3, and GPT-4o-mini. The detailed results are summarized in Table 2. Each experiment is conducted five times, and the results are presented as mean and standard deviation.

We train both traditional and deep learning-based baseline models on a subset of the available data. However, due to the high sparsity of the dataset, these models are unable to effectively learn meaningful patterns. As a result, we report the mean prediction as a reference point for comparison. The experimental results reveal a gradual decline in prediction accuracy across the Amazon, Goodreads, and Yelp datasets, reflecting an increasing level of recommendation difficulty. Baseline666 consistently outperforms other methods across nearly all settings, demonstrating strong robustness. Furthermore, Baseline666, DummyAgent, and RecHackers substantially outperform simpler designs,

Table 2: Performance comparison on classic recommendation tasks with the average HR@N metric (N=1,3,5).

| Category | Method | Amazon | | | Goodreads | | | Yelp | | |
|---|---|---|---|---|---|---|---|---|---|---|
| Traditional RS | MF | 30.1±1.2 | | | 14.1±0.3 | | | 34.4±0.9 | | |
| DL-based RS | LightGCN | 46.1±1.2 | | | 12.3±0.8 | | | 24.9±0.2 | | |
| | BC-Loss | 51.5±2.3 | | | 11.9±0.2 | | | 30.5±2.7 | | |
| | XsimGCL | 51.9±3.6 | | | 12.3±0.5 | | | 48.1±2.6 | | |
| | Model | Qwen | Deepseek | GPT | Qwen | Deepseek | GPT | Qwen | Deepseek | GPT |
| | BaseAgent | 13.8±2.6 | 21.3±1.3 | 11.8±1.5 | 17.2±1.2 | 20.7±7.7 | 16.2±2.1 | 12.6±1.1 | 15.5±0.5 | 16.6±0.5 |
| | CoTAgent | 13.9±1.4 | 19.4±1.7 | 12.6±1.5 | 17.1±0.4 | 21.2±2.1 | 17.3±0.9 | 13.3±0.7 | 17.7±0.7 | 16.7±0.2 |
| | MemoryAgent | 14.3±0.8 | 21.5±2.8 | 11.8±0.5 | 17.2±1.0 | 18.3±1.2 | 13.4±1.2 | 16.4±2.4 | 15.8±1.8 | 17.1±1.0 |
| Agentic RS | CoTMemAgent | 15.0±0.7 | 17.6±0.9 | 12.3±0.7 | 17.2±1.2 | 19.5±1.0 | 13.7±0.2 | 14.4±1.4 | 17.0±0.2 | 17.6±0.1 |
| | Baseline666 | 44.9±1.2 | 54.1±0.5 | 36.6±1.4 | 40.8±0.2 | 53.7±3.1 | 31.9±0.7 | 6.3±0.3 | 9.2±0.5 | 8.0±0.1 |
| | DummyAgent | 44.3±1.4 | 50.4±2.6 | 30.6±1.0 | 42.3±0.6 | 54.1±2.2 | 25.0±1.1 | 8.9±0.3 | 8.8±0.1 | 8.1±0.4 |
| | RecHackers | 48.3±4.4 | 55.1±0.6 | 40.3±2.3 | 45.3±0.6 | 52.4±1.5 | 27.0±5.0 | 8.8±0.6 | 9.3±0.5 | 6.0±1.5 |
| | Agent4Rec | 26.2±1.5 | 28.1±1.8 | 17.5±1.0 | 7.3±0.1 | 9.8±0.7 | 8.2±0.3 | 5.5±0.1 | 6.4±0.4 | 6.2±0.1 |

Table 3: Performance comparison on cold-start recommendation tasks with the average HR@N metric (N=1,3,5).

| Category | Method | Amazon | | Goodreads | | Yelp | |
|---|---|---|---|---|---|---|---|
| | | User | Item | User | Item | User | Item |
| Traditional RS | MF | 15.7±0.0 | 16.47±2.4 | 18.1±2.9 | 13.1±2.7 | 21.6±2.0 | 21.3 |
| DL-based RS | LightGCN | 12.8±0.7 | 15.5±2.2 | 16.7±1.8 | 13.9±1.6 | 21.5±2.8 | 14.5±2.1 |
| | BC-Loss | 12.7±0.7 | 17.9±2.3 | 14.4±1.1 | 15.1±3.2 | 21.7±1.9 | 14.9±2.1 |
| | XsimGCL | 12.4±0.7 | 14.2±1.1 | 16.4±2.0 | 11.4±2.5 | 20.1±3.1 | 14.2±1.8 |
| | BaseAgent | 15.3±1.1 | 14.3±1.1 | 18.2±0.3 | 15.5±1.0 | 3.1±0.2 | 2.7±0.2 |
| | CoTAgent | 15.9±0.2 | 13.3±0.4 | 17.9±0.7 | 16.1±1.4 | 2.0±0.1 | 4.3±0.4 |
| | MemoryAgent | 17.3±0.5 | 14.6±2.2 | 17.8±1.0 | 17.8±1.6 | 3.1±0.2 | 4.0±0.2 |
| | CoTMemAgent | 17.1±1.0 | 12.3±1.9 | 18.2±0.5 | 18.8±0.6 | 2.3±0.7 | 3.7±0.5 |
| Agentic RS | Baseline666 | 50.6±1.4 | 47.8±0.5 | 35.8±0.1 | 37.5±0.5 | 1.5±0.0 | 1.3±0.0 |
| | DummyAgent | 50.4±1.6 | 47.0±0.3 | 38.9±0.7 | 39.2±0.5 | 1.2±0.1 | 1.2±0.1 |
| | RecHackers | 52.4±0.2 | 52.1±1.9 | 38.8±0.4 | 40.1±1.0 | 1.4±0.1 | 0.9±0.0 |
| | Agent4Rec | 46.0±1.8 | 24.3±2.1 | 37.6±0.5 | 8.7±0.5 | 2.6±0.2 | 0.7±0.0 |

highlighting that a well-structured agent workflow can significantly enhance recommendation performance. These findings underscore the critical role of agent design in improving predictive accuracy within agent-based recommendation systems.

## 5.2 Performance on Cold-start Scenarios

In cold-start scenarios, we focus on users and items with limited historical interactions. Table 3 reports the performance of various methods based on Qwen-72B-Instruct. A comparison between Table 4 and Table 3 reveals a notable performance drop across nearly all baselines, highlighting the inherent uncertainty and challenge of cold-start settings. These results suggest that our proposed task can serve as a foundation for more comprehensive evaluations of recommendation efficiency under data-sparse conditions.

## 5.3 Performance on Evolving Interests

To assess models' adaptability to changing user preferences, we construct a scenario in which user interests evolve. This setting captures the temporal dynamics commonly observed in real-world recommendation tasks. The corresponding results are presented in Table 4. While most methods experience a performance decline under this setting, agent-based approaches such as Baseline666 and RecHackers demonstrate relatively stable performance, suggesting their enhanced ability to model and respond to evolving user behavior. We present a case study of Baseline666 in Figure 4.

Table 4: Performance comparison on evolving-interest recommendation tasks with the average HR@N metric (N=1,3,5).

| Category | Method | Amazon | | Goodreads | | Yelp | |
|---|---|---|---|---|---|---|---|
| | | Long Term | Short Term | Long Term | Short Term | Long Term | Short Term |
| Traditional RS | MF | 37.9±0.8 | 17.1±2.0 | 12.3±0.0 | 14.0±0.0 | 55.2±2.9 | 28.1±1.8 |
| DL-based RS | LightGCN | 55.1±1.8 | 15.3±2.2 | 16.0±0.0 | 14.0±0.0 | 54.7±1.1 | 27.2±1.9 |
| | BC-Loss | 53.7±3.8 | 14.0±2.0 | 16.0±0.0 | 14.7±0.0 | 56.34±3.2 | 24.3±2.1 |
| | XsimGCL | 31.3±2.7 | 11.1±2.4 | 16.0±0.0 | 14.7±0.0 | 45.0±3.9 | 25.1±0.9 |
| Agentic RS | BaseAgent | 14.2±1.0 | 16.4±1.5 | 28.7±0.0 | 28.3±0.5 | 4.7±0.2 | 5.5±0.4 |
| | CoTAgent | 13.3±0.9 | 16.8±0.5 | 25.1±0.2 | 26.8±1.7 | 4.3±0.4 | 5.7±0.2 |
| | MemoryAgent | 12.5±1.4 | 15.0±0.9 | 24.7±1.7 | 24.3±0.4 | 4.0±0.0 | 4.3±0.6 |
| | CoTMemAgent | 13.9±0.5 | 15.8±0.4 | 24.7±1.7 | 25.3±2.6 | 3.7±0.3 | 4.5±0.3 |
| | Baseline666 | 52.3±0.9 | 57.4±0.8 | 59.0±1.4 | 49.6±1.0 | 6.0±0.0 | 5.0±0.1 |
| | DummyAgent | 52.6±0.0 | 56.2±0.6 | 57.0±0.5 | 48.2±0.1 | 11.4±0.1 | 6.3±0.2 |
| | RecHackers | 54.4±2.2 | 59.2±0.6 | 58.7±1.2 | 52.3±1.2 | 10.5±0.1 | 6.6±0.5 |
| | Agent4Rec | 27.8±2.1 | 37.5±0.8 | 32.7±2.5 | 39.7±2.5 | 8.3±0.2 | 7.9±0.4 |

**Baseline666's core workflow**

```
You are a real user on an online platform. Your historical item review text
and stars are as follows: ['user_id', 'review_count', 'friends', 'stars'...]
Now you need to rank the following 20 items: [candidate item list]
'according to their match degree to your preference'.
Please rank the more interested items more front in your rank list.
The information of the above 20 candidate items is as follows: ['item_id',
'name', 'stars', 'review_count', 'attributes', 'title','average_rating',
'rating_number', 'description'...]
Your final output should be ONLY a ranked item list of [Candidate List]
with the following format!
DO NOT introduce any other item ids! DO NOT output your analysis process!
The correct output format: [Sorted Candidate Item List]
```

Figure 4: Illustration of the agentic workflow of a superior recommendation agent (Baseline666), which conducts domain-adaptive item-side feature engineering to enhance personalization.

# 6 Related Works

## 6.1 Agentic Recommender Systems

Recent advances in LLM-based agents have introduced transformative paradigms for recommender systems. These agentic systems demonstrate autonomous capabilities to collect and process user-item interactions while leveraging sophisticated reasoning for personalized recommendations. Existing approaches can be categorized into three main branches: (1) ranking-oriented agents [26–28] that infer user preferences from historical behavior to generate recommendations, exemplified by Rec-Mind [26]; (2) simulation-oriented agents [12, 14, 29] that simulate human-like behavior through role-playing, such as Agent4Rec [12]; and (3) interactive conversational agents [13, 30, 31] that frame recommendation as dialogue-based intent understanding, as demonstrated by RAH [13].

However, the evaluation of these prior works differ significantly in focus from our benchmark. For instance, Agent4Rec [12] primarily assesses generative agents for user behavior simulation. Its focus is on the fidelity of an agent's role-playing (e.g., rating, commenting) against real user behavior, not on the agent's capacity for autonomous information gathering to make recommendations. AgentCF [14] proposes a multi-agent simulation approach for collaborative filtering, modeling users and items as agents. It emphasizes personalized behavior modeling within the simulation, but doesn't evaluate an agent's ability to execute an end-to-end recommendation task. AFL [32] primarily emphasizes the agent's ability to learn from user feedback. Our work, in contrast, focuses on the agent's information retrieval and personalized analysis capabilities, which are more aligned with real-world recommendation scenarios.

## 6.2 LLM-based Agents

Recent advances in LLMs have enabled the development of autonomous agents capable of sophisticated reasoning [25, 18, 33], human-like simulation [16, 34], and decision-making in diverse domains such as gaming [35] and web navigation [36]. Beyond single-agent applications, multi-agent systems [37–39] are also being explored for their emergent collaborative intelligence. A key direction in this field is enhancing agents with personalization capabilities, often achieved through two primary technical approaches. The first is building sophisticated memory mechanisms [40–42] that allow agents to continuously adapt to user contexts. The second is human preference alignment [43], which involves fine-tuning models to better match individual user preferences. While these techniques are promising for recommendation agents, evaluating personalization remains a challenge. For instance, PREFEVAL [44] benchmark assesses an LLM's ability to infer user preferences from static instructions. In contrast, our work evaluates agents in dynamic, interactive scenarios, focusing on their ability to autonomously collect information for personalization—a critical distinction from existing evaluation methods.

## 6.3 LLM Agent Benchmarks

Several benchmarks have been developed to evaluate the general capabilities of LLM agents [45–49]. AgentBench [46] focuses on reasoning and decision-making in multi-turn, open-ended settings across code, game, and web environments. AgentBoard [47] encompasses nine diverse tasks, emphasizing multi-turn interaction and long-term task execution. TheAgentCompany [48] assesses agents on human work-related tasks like software engineering and financial analysis. While our work also evaluates autonomous agents interacting with an environment, our domain is entirely distinct. The aforementioned benchmarks primarily evaluate general-purpose agent capabilities. In contrast, AgentRecBench is the first evaluation framework specifically designed for LLM agents in personalized recommendation tasks. Our framework uniquely assesses an agent's ability for personalized preference modeling, personalized information retrieval, and end-to-end recommendation task execution, which are not covered by existing general-purpose benchmarks.

## 7 Conclusions

In this work, we present AgentRecBench, the first comprehensive benchmark for evaluating the emerging LLM-powered agentic recommender systems. The benchmark establishes rigorous evaluation protocols across multiple domains and scenarios, supported by our novel textual interaction environment that integrates three rich recommendation datasets. Through extensive empirical analysis of 10+ classical and agentic methods, we not only demonstrate the superior performance of LLM-based approaches but also identify their critical designs. The proposed modular agent design framework and standardized evaluation platform provide researchers with essential tools to advance the development of agentic recommender systems. We believe this work offers a fundamental platform for advancing the next-generation recommender systems.

## Acknowledgments and Disclosure of Funding

This work was supported in part by the National Natural Science Foundation of China (No. 62472241, No. 23IAA02114).

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

# A  Appendix

## A.1  Broader impacts

The benchmark and toolkit presented in this work are expected to advance the development of agentic recommender systems and bring significant positive societal implications. The improved contextual understanding of agentic recommender systems enables higher-quality personalized recommendations that can benefit diverse domains, including e-commerce, short-video platforms, etc. Furthermore, the standardized evaluation framework facilitates the development of more robust and adaptable recommendation systems that can better serve evolving user needs. Our platform can also provide a flexible foundation for incorporating ethical AI principles, such as fairness and privacy preservation, into future recommendation systems. These advancements collectively contribute to building more user-centric, explainable, and socially beneficial recommendation technologies.

## A.2  Limitations

While AgentRecBench provides a comprehensive evaluation framework for agentic recommender systems, several limitations point to valuable directions for future work. First, the current benchmark primarily focuses on evaluating emerging agentic recommender systems, and we plan to incorporate more traditional and deep learning-based baselines for more thorough comparative analysis. Second, our environment currently operates on textual information, and we aim to extend it to incorporate multimodal data (e.g., images, videos) to better reflect real-world recommendation scenarios where agents need to process diverse content types. Finally, while the current framework evaluates single-agent systems, we plan to extend it to support the evaluation of multi-agent recommendation systems where collaborative or competitive agents interact to improve recommendation outcomes.

## A.3  Potential application scenarios of the benchmark

Recommendation agents offer significant advantages over traditional recommendation methods, which typically rely on platform-centric training with pre-processed user historical data for content distribution. Agents, in contrast, excel by autonomously collecting and analyzing information and flexibly adjusting recommendation strategies in scenarios where data is scattered or not pre-defined. This capability highlights two promising application scenarios. First, the realization of **personalized user-side recommendation assistants** is compelling: a dedicated agent continuously tracks a user's behavior and feedback, dynamically adjusting strategies to adapt to evolving interests and even proactively anticipating needs. This shifts the paradigm from platform-centric to a user-centric, dynamic, and highly personalized experience. Second, agents are crucial for enhancing **cold-start recommendations**. Their powerful autonomous information collection and semantic analysis capabilities allow them to rapidly build profiles for new users or items, significantly improving recommendation effectiveness in data-scarce situations where traditional methods struggle, a potential which our experimental results also validate.

## A.4  Supplemented experimental results of the cold-start recommendation tasks and evolving-interest recommendation tasks

Tables 5 and 6 show the complementary experimental results of our task on DeepSeek V3 and GPT-4o-mini. Overall, our evaluation results on other mainstream models, such as DeepSeek V3 and GPT-4o-mini, demonstrate that the agent's performance is consistent with the findings presented in the main text. Baseline666, DummyAgent, and RecHackers all exhibit strong performance, further validating the robustness and reliability of our evaluation method in accurately reflecting agent capabilities.

## A.5  Empirical comparison of traditional evaluation frameworks and autonomous agent evaluation in AgentRecBench

Autonomous agents inherently require dynamic interaction with their environment, often involving multi-step reasoning and adaptive tool calling. Traditional evaluation frameworks based on static data (e.g., user-item matrices) or fixed LLM prompts inherently lack this interactive capability. To further illustrate this, we experiment with providing LLMs with traditional static user-item matrix

Table 5: Performance comparison on cold-start recommendation tasks with the average HR@N metric (N=1,3,5).

**DeepSeek-V3**

| Category | Method | Amazon | | Goodreads | | Yelp | |
|---|---|---|---|---|---|---|---|
| | | User | Item | User | Item | User | Item |
| | BaseAgent | 16.0±1.9 | 15.1±1.5 | 20.4±0.6 | 21.1±2.2 | 3.8±0.1 | 4.4±0.1 |
| | CoTAgent | 19.4±0.4 | 13.6±2.0 | 17.8±3.1 | 17.0±3.4 | 4.5±0.0 | 4.3±0.1 |
| | MemoryAgent | 17.4±0.6 | 13.5±1.8 | 19.0±1.1 | 22.6±4.1 | 4.2±0.8 | 4.1±0.1 |
| | CoTMemAgent | 18.3±3.2 | 15.0±2.3 | 14.8±1.5 | 17.3±0.4 | 3.6±0.0 | 3.8±0.4 |
| Agentic RS | Baseline666 | 55.7±1.6 | 54.1±3.3 | 41.0±2.8 | 50.3±2.9 | 1.1±0.1 | 2.9±0.0 |
| | DummyAgent | 54.2±0.6 | 51.7±0.3 | 43.6±5.3 | 49.3±1.4 | 1.5±0.1 | 2.8±0.1 |
| | RecHackers | 59.5±0.9 | 56.9±0.2 | 46.8±0.9 | 46.1±2.2 | 2.8±0.1 | 4.0±0.2 |
| | Agent4Rec | 48.1±1.4 | 32.7±1.1 | 35.9±0.2 | 11.1±0.2 | 1.7±0.2 | 0.8±0.0 |

**GPT-4o-mini**

| Category | Method | Amazon | | Goodreads | | Yelp | |
|---|---|---|---|---|---|---|---|
| | | User | Item | User | Item | User | Item |
| | BaseAgent | 14.8±1.5 | 8.8±0.5 | 20.3±0.9 | 16.2±1.1 | 3.0±0.0 | 4.4±0.0 |
| | CoTAgent | 15.5±0.0 | 8.7±0.2 | 20.4±0.1 | 13.9±0.3 | 3.0±0.0 | 4.3±0.0 |
| | MemoryAgent | 15.3±3.2 | 7.4±0.5 | 15.8±0.1 | 17.7±3.0 | 2.8±0.1 | 3.5±0.1 |
| | CoTMemAgent | 15.8±0.8 | 7.4±0.2 | 16.8±0.2 | 17.6±0.4 | 3.0±0.0 | 4.1±0.0 |
| Agentic RS | Baseline666 | 44.5±0.0 | 31.7±0.2 | 35.3±0.4 | 27.7±0.6 | 0.3±0.0 | 3.1±0.0 |
| | DummyAgent | 44.3±0.5 | 29.6±0.4 | 37.1±0.0 | 26.1±1.3 | 0.3±0.0 | 3.7±0.0 |
| | RecHackers | 46.8±10.5 | 39.3±8.5 | 40.4±1.4 | 31.1±2.2 | 1.0±0.1 | 2.9±0.1 |
| | Agent4Rec | 34.9±1.5 | 20.6±1.4 | 38.7±0.7 | 7.9±0.1 | 0.7±0.0 | 0.8±0.0 |

Table 6: Performance comparison on evolving-interest recommendation tasks with the average HR@N metric (N=1,3,5).

**DeepSeek-V3**

| Category | Method | Amazon | | Goodreads | | Yelp | |
|---|---|---|---|---|---|---|---|
| | | Long Term | Short Term | Long Term | Short Term | Long Term | Short Term |
| | BaseAgent | 21.7±0.7 | 21.7±0.7 | 33.2±1.6 | 33.2±1.6 | 5.2±0.0 | 5.2±0.0 |
| | CoTAgent | 19.1±1.2 | 19.1±1.2 | 25.6±0.7 | 25.6±0.7 | 4.3±0.0 | 4.3±0.0 |
| | MemoryAgent | 21.1±2.2 | 21.1±2.2 | 29.3±4.6 | 29.3±4.6 | 4.5±0.3 | 4.5±0.0 |
| | CoTMemAgent | 16.7±0.2 | 16.7±0.2 | 22.8±3.3 | 22.8±3.3 | 4.2±0.0 | 4.2±0.0 |
| Agentic RS | Baseline666 | 61.0±0.4 | 61.0±0.4 | 66.6±0.7 | 66.6±0.7 | 5.2±0.0 | 5.2±0.0 |
| | DummyAgent | 61.3±0.6 | 61.3±0.6 | 65.2±2.5 | 65.2±2.4 | 10.6±0.1 | 10.6±0.1 |
| | RecHackers | 66.2±0.4 | 66.2±0.4 | 68.4±1.2 | 68.4±1.2 | 10.7±0.1 | 10.7±0.1 |
| | Agent4Rec | 32.5±1.8 | 32.5±1.8 | 41.5±0.1 | 41.5±0.1 | 10.3±0.2 | 10.3±0.2 |

**GPT-4o-mini**

| Category | Method | Amazon | | Goodreads | | Yelp | |
|---|---|---|---|---|---|---|---|
| | | Long Term | Short Term | Long Term | Short Term | Long Term | Short Term |
| | BaseAgent | 11.7±0.2 | 13.0±0.1 | 13.7±0.2 | 14.4±0.7 | 4.6±0.3 | 4.5±0.1 |
| | CoTAgent | 11.8±0.3 | 13.0±0.4 | 13.9±0.2 | 14.3±0.4 | 5.1±0.0 | 4.8±0.1 |
| | MemoryAgent | 11.7±0.1 | 12.6±0.2 | 17.3±1.7 | 16.3±1.0 | 3.6±0.4 | 4.8±0.1 |
| | CoTMemAgent | 11.8±0.1 | 12.6±0.2 | 18.0±0.3 | 15.8±0.4 | 5.2±0.1 | 4.8±0.1 |
| Agentic RS | Baseline666 | 37.9±0.3 | 48.1±2.2 | 47.8±2.2 | 48.1±0.5 | 5.8±0.1 | 5.8±0.1 |
| | DummyAgent | 37.8±2.6 | 43.8±2.6 | 43.1±0.5 | 42.1±0.6 | 3.8±0.5 | 2.1±0.1 |
| | RecHackers | 43.8±4.4 | 54.3±3.5 | 53.4±1.2 | 53.2±3.8 | 4.7±0.9 | 3.0±0.4 |
| | Agent4Rec | 23.0±0.9 | 35.5±1.7 | 30.9±2.7 | 36.0±1.4 | 5.8±0.4 | 4.2±0.3 |

| Method | Qwen 2.5 | Deepseek V3 | GPT-4o-mini |
|---|---|---|---|
| LLM with static matrix data | 19.7 | 9.0 | 15.3 |
| Autonomous Rec Agent (Baseline666) | 44.9 | 54.1 | 36.6 |

Table 7: Performance comparison of agentic recommender systems on the traditional evaluation framework and autonomous agent evaluation.

data for recommendation tasks on Amazon data. As shown in Table 7, performance was significantly lower compared to agents that dynamically collect information. This demonstrates that static, non-interactive data fails to fully leverage an agent's ability for autonomous information gathering and personalized reasoning, and thus is inadequate for agentic recommendation evaluation.

## A.6 Case Study

We present the core workflows of the two agents, DummyAgent, and RecHackers in Figure 5, and 6, respectively. Overall, these agents rely on similar types of information for the ranking process, although their specific implementations differ. The ranking decisions are primarily based on three key components: (1) historical user reviews, which reflect past preferences; (2) a list of candidate items to be ranked; and (3) detailed item information, which helps evaluate the relevance of each item to the user's preferences.

A major innovation among these agents lies in item-side feature engineering. Notably, the Baseline666 team employed platform-specific feature extraction methods, enabling a robust and adaptable ranking strategy across different data sources. For instance, on the Amazon platform, features such as item ID, name, star rating, number of reviews, and item description were extracted. On Yelp, they focused on core attributes like item ID, name, star rating, and review count. In contrast, the Goodreads platform required more diverse features, including author, publication year, and similar books.

Review-side feature engineering represents another critical component, aimed at identifying and extracting the most informative reviews to enrich the understanding of both user preferences and item characteristics. For example, the DummyAgent team implemented a platform-tailored strategy: on Yelp, they extracted not only the review text but also interactive attributes such as "useful," "cool," and "funny"; on Amazon, they incorporated publication dates and purchase verification indicators; and on Goodreads, in addition to the review text and ratings, they utilized metadata such as review date, number of votes, number of comments, and reading status.

In summary, the key design elements across these agents can be distilled into three core principles: (1) effective workflows are built on a combination of user history, candidate items, item details, and platform-specific features, all integrated through large language models (LLMs) to produce rankings; (2) extracting representative, platform-specific item attributes is essential for enhancing model performance; and (3) prioritizing reviews that are both highly relevant and information-rich is crucial for improving ranking quality.

```
You are a real user on an online platform. Your historical item review text
and stars are as follows: ['item_id', 'text', 'timestamp', 'source',
'type', 'sub_item_id', 'stars', 'helpful_vote', 'verified_purchase',
'title'...]. Now you need to rank the following {final_item_list_len}
items: {final_item_list} according to their match degree to your preference.
Please rank the more interested items more front in your rank list. The
information of the above {final_item_list_len} candidate items is as
follows: ['item_id', 'name', 'stars', 'review_count', 'attributes',
'title', 'average_rating', 'rating_number', 'description', 'ratings_count',
'title_without_series'].
Your final output should be ONLY a ranked item list of {final_item_list}
with the following format, DO NOT introduce any other item ids!
DO NOT output your analysis process! The correct output format: [Sorted
Candidate Item List]
```

Figure 5: Illustration of the agentic workflow of a superior recommendation agent (DummyAgent), which conducts domain-adaptive item-side feature engineering to enhance personalization.

```
'You are a real human user on {task_type}, a platform for crowd-sourced
{task_item} reviews'. Here is your {task_type} profile and review history:
{user}. Your historical {task_item} reviews show your preference as
follows: ['user_id', 'review_count', 'friends', 'stars'...].
Now you need to rank the following 20 {task_item}: [Candidate List]
according to their match degree to your preference. The information of the
above 20 candidate {task_item} is as follows: ['item_id', 'name', 'stars',
'review_count', 'attributes', 'title', 'average_rating', 'rating_number',
'description', 'ratings_count'...]. Your final output should be ONLY a
ranked {task_item} list of [Candidate List] with the following format, DO
NOT introduce any other {task_item} ids!
You only need to select the top 5 {task_item} from the candidate list.
'Please rank the more interested {task_item} more front in your rank list.'
You should think step by step before your final answer. DO NOT output your
analysis process! Follow the correct final answer output format strictly,
remember to output {task_item} ids instead of {task_item} names:
[Sorted Candidate Item List]
```

Figure 6: Illustration of the agentic workflow of a superior recommendation agent (RecHacker), which conducts domain-adaptive item-side feature engineering to enhance personalization.

