# OpenReview forum: "AgentRecBench: Benchmarking LLM Agent-based Personalized Recommender Systems"
_NeurIPS.cc/2025/Datasets_and_Benchmarks_Track — NeurIPS 2025 Datasets and Benchmarks Track spotlight_

### Official Review · Reviewer_FmG6 · 2025-06-11

**Rating:** 5
**Confidence:** 3

**Summary:**

AgentRecBench is the first large-scale benchmark designed to systematically evaluate agentic recommender systems powered by LLMs alongside traditional methods across diverse recommendation scenarios. It features an interactive simulator with rich user/item metadata and three key tasks—classic, evolving-interest, and cold-start recommendations. The benchmark provides a unified framework and comprehensive comparisons of 10 methods, demonstrating the superior performance of agentic systems and offering design insights to guide future development.

**Dataset Code Accessibility:**

Yes

**Ethical Considerations:**

No, there are no or only very minor ethics concerns

**Final Justification:**

The authors have addressed my comments. I maintain my positive score. Thank you.

**Limitations Weaknesses:**

1. How is the quality of the constructed dataset ensured? Does the dataset include any human verification? Are there noisy or irrelevant data that have been filtered out from sources such as Yelp, Goodreads, and Amazon?

2. It would be helpful to conclude with some potential application scenarios of this benchmark, including concrete examples, to provide a more intuitive understanding of its practical use and impact.

3. Discussing related techniques in personalized AI assistants ([1-5]) and existing benchmarks [6] would help provide a broader perspective on recommending user-centric services, e.g.,

[1] Memorybank: Enhancing Large Language Models with Long-Term Memory (AAAI 2024)

[2] Crafting Personalized Agents through Retrieval-Augmented Generation on Editable Memory Graphs (EMNLP 2024)

[3] LLM-Personalize: Aligning LLM Planners with Human Preferences via Reinforced Self-Training for Housekeeping Robots (COLING 2025)

[4] Memory-Augmented LLM Personalization with Short- and Long-Term Memory Coordination (NAACL 2024)

[5] CogAgent: A Visual Language Model for GUI Agents (CVPR 2024)

[6] Do LLMs Recognize Your Preferences? Evaluating Personalized Preference Following in LLMs (ICLR 2025)

Incorporating these works would enrich the discussion and highlight current trends and challenges in LLM-based Agents.

**Strengths Contributions:**

1. AgentRecBench is the first large-scale benchmark comparing agentic and traditional recommender systems across diverse scenarios.

2. It offers a textual simulator with multi-domain datasets and a unified agent development framework for rapid prototyping and evaluation.

3. The analysis of 10 methods reveals key design insights and provides practical guidelines for building better agentic recommender systems.

---

> ### Author Rebuttal · Authors · 2025-07-31
>
> **Q1: How is the quality of the constructed dataset ensured? Does the dataset include any human verification? Are there noisy or irrelevant data that have been filtered out from sources such as Yelp, Goodreads, and Amazon?**
>
> **Response:** To ensure the quality of the benchmark, we adopted a multi-faceted quality control strategy and conducted human verification, including scenario-based data selection, dynamic data visibility control and empirical validation via community challenge.
>
> - **Scenario-based data selection**: Each recommendation scenario (e.g., cold-start, evolving interests) utilizes carefully tailored filters to ensure data relevance and minimize noise. For instance, cold-start scenarios focus on users or items with limited historical interactions, while evolving-interest tasks rely on time-segmented user histories. We also manually evaluated and selected the data collection time window with higher interaction density to ensure richer and more informative datasets. This manual curation step acts as a form of human oversight on data quality for critical scenarios.
>
> - **Dynamic data visibility control**: As detailed in Section 3.2, our "Dynamic Data Visibility Control" mechanism employs time- and interaction-based thresholds to filter out inherently noisy or sparse data from the Yelp, Amazon, and Goodreads sources. This includes criteria such as minimum review counts and temporal constraints, ensuring that only sufficiently informative and representative data is visible to agents for robust recommendation modeling.
>
> - **Empirical validation via community challenge**: The dataset's practical quality and effectiveness were rigorously validated through the AgentSociety Challenge. This challenge attracted 295 teams and over 1,400 submissions, demonstrating the dataset's utility and reliability in real-world agent-based recommendation tasks. The successful deployment and widespread adoption of the dataset within such a competitive environment serve as strong empirical evidence of its quality and fitness for purpose.
>
>
>
> **Q2: It would be helpful to conclude with some potential application scenarios of this benchmark, including concrete examples, to provide a more intuitive understanding of its practical use and impact.**
>
> **Response:** Thanks for the valuable suggestion. Our benchmark is specifically designed for recommendation agents, which offer significant advantages over traditional recommendation methods. Traditional approaches typically rely on platforms to train and maintain models based on pre-processed user historical data for personalized content distribution. Recommendation agents, however, excel in **scenarios where information is scattered or not pre-defined**, where they can autonomously collect and analyze information, and flexibly adjust recommendation strategies. Here are two specific promising application scenarios for recommendation agents:
>
> - **Personalized User-Side Recommendation Assistants**: One compelling application is the realization of personalized user-side recommendation assistants. Imagine each user having a dedicated recommendation agent within content platforms. This agent would continuously track the user's behavior and feedback on the platform, dynamically adjusting recommendation strategies to adapt to evolving user interests. It could even proactively anticipate user needs, offering personalized recommendations before an explicit query is made. This shifts the paradigm from platform-centric content distribution to **user-centric, dynamic, and highly personalized recommendation experiences**.
>
> - **Enhancing Cold-Start Recommendations**: Another crucial application lies in improving cold-start scenarios. Recommendation agents, with their powerful capabilities for autonomous information collection and semantic analysis, can rapidly build profiles for new users or new items. This significantly enhances the effectiveness of recommendations in cold-start situations, where traditional methods struggle due to a lack of historical data. Our experimental results (Table 3) also validate the potential of agents in this domain.
>
>
>
> **Q3: Discussing related techniques in personalized AI assistants ([1-5]) and existing benchmarks [6] would help provide a broader perspective on recommending user-centric services,e.g, ..... Incorporating these works would enrich the discussion and highlight current trends and challenges in LLM-based Agents.**
>
> **Response:** We appreciate the provided relevant works. We discuss the mentioned personalized AI assistants and related benchmarks as follows and will expand our Introduction part to include the discussion.
> Similar to our focus, personalized AI assistants are LLM-based agents deeply integrated with personal data for assistance in both digital (e.g., CogAgent [5]) and physical worlds (e.g., LLM-Personalize [3]). Technically, personalization is enhanced through:
>
> - **Memory Mechanisms**: Works like MemoryBank [1] and EMG-RAG [2] design sophisticated memory systems for personalized responses, continuously adapting to user personalities and contexts. Similarly, [4] uses a bionic memory for medical assistants. These memory designs are also promising for enhancing recommendation agents.
> - **Human Preference Alignment**: Another approach involves fine-tuning LLM models using human preferences to achieve personalization [3].
>
> In terms of benchmarks, PREFEVAL [6] assess LLMs' ability to infer and follow personalized user preferences, however, it only includes **static instruction datasets**. Our work, in contrast, evaluates personalized recommendation agents that interact autonomously with environments and actively collect information for personalization. This focus on dynamic, interactive agent behavior in personalized recommendation scenarios is what uniquely distinguishes our work.

---

### Official Review · Reviewer_yMHW · 2025-06-25

**Rating:** 5
**Confidence:** 4

**Summary:**

This paper presents AgentRecBench, a comprehensive benchmarking framework specifically designed for LLM-based agentic recommender systems. Through systematic experimental design and rigorous evaluation methodologies, this work addresses the critical gap of lacking standardized evaluation protocols in the field of LLM-powered recommendation agents.

This work provides open and complete community resources. The benchmark's practicality was validated through the AgentSociety Challenge open competition. The research team maintains a public leaderboard and provides full access to the benchmarking environment, effectively promoting reproducible research and collaborative development in this field

The key innovations are manifested in three aspects:
- Multi-scenario evaluation environment
- Modular agent development framework
- Comprehensive performance evaluation

**Additional Feedback:**

1. The authors could conduct multiple runs for individual models (even with data subsets) to quantify variance
2. The authors should include more modern non-agentic baselines to strengthen comparative analysis. If older methods remain widely used benchmarks, justify their selection
3. The authors should add a dedicated discussion on how AgentRecBench complements/distinguishes from existing agent benchmarks. And they should emphasize unique contributions (e.g., recommendation focus) to clarify positioning

**Dataset Code Accessibility:**

Yes

**Dataset Code Comments:**

Open-source, HuggingFace dataset link provided in Abstract for easy access

**Ethical Comments:**

The datasets are derived from public data without involving crowdsourcing or human subjects, complying with ethical standards.

**Ethical Considerations:**

No, there are no or only very minor ethics concerns

**Final Justification:**

After carefully reviewing the authors' rebuttal and the additional materials they provided, I am satisfied with their thorough responses to my concerns. Specifically:
1) Reproducibility & Stability: The authors have conducted five independent experiments to address the instability of LLM outputs and reported standard deviations, significantly improving the reliability of their results.
2) Updated Baselines: They have added comparisons with two recent state-of-the-art methods (published in top-tier venues), further validating the competitiveness of their approach.
3) Related Work Expansion: The discussion now includes a more comprehensive analysis of similar LLM-based agent research, better situating their work in the broader literature.

Given these substantial improvements, I have decided to raise my rating from 4 to 5.

**Limitations Weaknesses:**

1. Uncertainty in LLM-based results and insufficient statistical significance analysis: As noted in the checklist (Section 7), due to API cost constraints, the experiments lacked repeated trials or error bar calculations, instead validating performance across three LLM architectures (Qwen, DeepSeek, GPT). While this demonstrates cross-model consistency, it doesn't account for inherent randomness in single-model outputs (e.g., LLM stochastic sampling).
2. Potentially outdated baseline comparisons: The selected traditional baselines (e.g., 2009 Matrix Factorization and 2020 LightGCN), while seminal, may not fully represent recent advances in non-agentic recommender systems.
3. Incomplete coverage of related work on LLM agent benchmarks: The related work section doesn't discuss existing LLM agent benchmarking frameworks like AgentBench, AgentBoard (NeurIPS'24), or TheAgentCompany. While application domains differ, methodological similarities exist. Addressing these could better highlight AgentRecBench's novelty.

**Strengths Contributions:**

1. Novel dataset construction approach: By curating and extending existing datasets, the study expands their application scope while avoiding privacy or feasibility issues associated with crowdsourcing.
2. Practical value of datasets: The datasets are open-source and have been successfully applied in real-world scenarios.
3. Clear differentiation between agentic recommender systems and traditional methods (e.g., matrix factorization, LightGCN), highlighting advantages such as interpretability and dynamic adaptability, while emphasizing the lack of standardized benchmarks to justify its contribution.
4. The paper is well-structured with clear language and appropriate visual content

---

> ### Author Rebuttal · Authors · 2025-07-31
>
> **Q1: Uncertainty in LLM-based results and insufficient statistical significance analysis: As noted in the checklist (Section 7), due to API cost constraints, the experiments lacked repeated trials or error bar calculations, instead validating performance across three LLM architectures (Qwen, DeepSeek, GPT). While this demonstrates cross-model consistency, it doesn't account for inherent randomness in single-model outputs (e.g., LLM stochastic sampling).**
>
> **Response:** Thanks for this insightful feedback. We have conducted five independent runs for all experiments in Table 2 and reported results with standard deviations as follows. The full set of results (covering classic, cold-start, and evolving-interest tasks) will be added in the revised version of the paper.
>
> | Method / Dataset | Amazon (Qwen) | Amazon (DeepSeek) | Amazon (GPT) | Goodreads (Qwen) | Goodreads (DeepSeek) | Goodreads (GPT) | Yelp (Qwen) | Yelp (DeepSeek) | Yelp (GPT) |
> | ---------------- | ------------- | ----------------- | ------------ | ---------------- | -------------------- | --------------- | ----------- | --------------- | ---------- |
> | BaseAgent        | 13.8±2.6      | 21.3±1.3          | 11.8±1.5     | 17.2±1.2         | 20.7±7.7             | 16.2±2.1        | 12.6±1.1    | 15.5±0.5        | 16.6±0.5   |
> | CoTAgent         | 13.9±1.4      | 19.4±1.7          | 12.6±1.5     | 17.1±0.4         | 21.2±2.1             | 17.3±0.9        | 13.3±0.7    | 17.7±0.7        | 16.7±0.2   |
> | MemoryAgent      | 14.3±0.8      | 21.5±2.8          | 11.8±0.5     | 17.2±1.0         | 18.3±1.2             | 13.4±1.2        | 16.4±2.4    | 15.8±1.8        | 17.1±1.0   |
> | CoTMemAgent      | 15.0±0.7      | 17.6±0.9          | 12.3±0.7     | 17.2±1.2         | 19.5±1.0             | 13.7±0.2        | 14.4±1.4    | 17.0±0.2        | 17.6±0.1   |
> | Baseline666      | 44.9±1.2      | 54.1±0.5          | 36.6±1.4     | 40.8±0.2         | 53.7±3.1             | 31.9±0.7        | 6.3±0.3     | 9.2±0.5         | 8.0±0.1    |
> | DummyAgent       | 44.3±1.4      | 50.4±2.6          | 30.6±1.0     | 42.3±0.6         | 54.1±2.2             | 25.0±1.1        | 8.9±0.3     | 8.8±0.1         | 8.1±0.4    |
> | RecHackers       | 48.3±4.4      | 55.1±0.6          | 40.3±2.3     | 45.3±0.6         | 52.4±1.5             | 27.0±5.0        | 8.8±0.6     | 9.3±0.5         | 6.0±1.5    |
> | Agent4Rec        | 26.2±1.5      | 28.1±1.8          | 17.5±1.0     | 7.3±0.1          | 9.8±0.7              | 8.2±0.3         | 5.5±0.1     | 6.4±0.4         | 6.2±0.1    |
>
>
>
> **Q2: Potentially outdated baseline comparisons: The selected traditional baselines (e.g., 2009 Matrix Factorization and 2020 LightGCN), while seminal, may not fully represent recent advances in non-agentic recommender systems.**
>
> **Response:** Thanks for the valuable feedback. We have incorporated two recent and strong traditional recommendation methods: **BC-Loss** [1] and **XSimGCL** [2], as additional baselines. The updated results will be integrated into the experimental results section of the revised paper.
>
> |                 | Amazon  | Amazon  | Goodreads | Goodreads | Yelp    | Yelp    |
> | --------------- | ------- | ------- | --------- | --------- | ------- | ------- |
> |                 | BC_Loss | XSimGCL | BC_Loss   | XSimGCL   | BC_Loss | XSimGCL |
> | classic         | 53.7    | 49.3    | 14.7      | 14.3      | 25.7    | 47.0    |
> | user_cold_start | 17.0    | 15.0    | 17.0      | 17.7      | 22.7    | 20.0    |
> | item_cold_start | 17.3    | 14.7    | 13.2      | 14.0      | 11.3    | 11.7    |
> | long_term       | 51.3    | 32.3    | 12.3      | 12.3      | 53.3    | 45.3    |
> | short_term      | 13.7    | 6.0     | 14.0      | 14.0      | 25.3    | 23.0    |
>
> [1]Zhang, An, et al. "Incorporating bias-aware margins into contrastive loss for collaborative filtering." NeurIPS 2022.
>
> [2]Yu, Junliang, et al. "XSimGCL: Towards Extremely Simple Graph Contrastive Learning for Recommendation." TKDE 2024.
>
>
>
> **Q3: Incomplete coverage of related work on LLM agent benchmarks: The related work section doesn't discuss existing LLM agent benchmarking frameworks like AgentBench, AgentBoard (NeurIPS'24), or TheAgentCompany. While application domains differ, methodological similarities exist. Addressing these could better highlight AgentRecBench's novelty.**
>
> **Response:** Thanks for this suggestion. We now include a concise discussion of some related LLM-agent benchmarks and will revise it in Introduction part:
>
> - **AgentBench** focuses on evaluating general reasoning and decision-making abilities in multi-turn, open-ended generation settings across code, game, and web environments.
> - **AgentBoard**  encompasses nine diverse tasks, from embodied AI and game agents to web and tool agents, emphasizing general-purpose agents' multi-turn interaction and long-term task execution, with support for fine-grained task progress evaluation.
> - **TheAgentCompany**  assesses agents' performance on human work-related tasks like software engineering, project management, and financial analysis, requiring web browse, coding, and interaction with simulated co-workers for long-term collaborative tasks.
>
> Our work also evaluates autonomous agents interacting with an environment, focusing on their complex reasoning and autonomous tool-use capabilities. However, our domain is entirely distinct. The aforementioned existing works primarily evaluate general-purpose agent capabilities. AgentRecBench is the first evaluation framework specifically designed for LLM agents in personalized recommendation tasks. Our framework uniquely assesses an agent's ability for personalized preference inference, personalized information retrieval, and end-to-end recommendation task execution, which are not covered by existing general-purpose benchmarks.

---

### Official Review · Reviewer_op5v · 2025-07-02

**Rating:** 4
**Confidence:** 5

**Summary:**

Traditional recommender systems face limitations in interpretability, contextual adaptation, and dynamic strategy optimization. While LLM-based agent recommenders show promise, the field lacks standardized evaluation protocols.

1.Introduces AgentRecBench, the first unified benchmark for evaluating LLM agent-based recommenders alongside traditional methods across diverse scenarios (classic, cold-start, evolving interests).

2.Designs an interactive simulator integrating multi-domain datasets (Yelp, GoodReads, Amazon) with: Dynamic data visibility control (time/item filters)， sandardized query interfaces for agent-environment interaction.

3.Proposes a flexible framework with four core modules: Dynamic planning, complex reasoning and tool utilization.

4.Evaluates 10+ methods (traditional, DL-based, agentic) across 3 datasets.

**Dataset Code Accessibility:**

Yes

**Ethical Considerations:**

No, there are no or only very minor ethics concerns

**Final Justification:**

The authors have provided a comprehensive review which address most of my concerns.

**Limitations Weaknesses:**

1.The authors are encouraged to elaborate on the challenges in constructing the benchmark (such as modeling environmental dynamics and designing multi-scenario evaluation metrics), and to highlight how this work complements limitations of existing evaluation frameworks (e.g., RecSys challenges) to strengthen its persuasiveness.

2.The introduction emphasizes this work as the “first comprehensive benchmark,” but does not compare it with similar attempts (such as the evaluation in Agent4Rec). A differential analysis is needed to clarify distinctions.

3.The evolution of traditional recommendation methods (e.g., MF, LightGCN) is described only briefly; the authors should clarify why existing evaluations are inadequate for agents (for example, due to the lack of interactive decision-making).

4.The paper does not explain how “dynamic planning” is implemented in terms of module interactions (such as the triggering conditions for task decomposition).

5.The memory management mechanism (such as experience storage and retrieval strategies) lacks implementation details.

6.The performance of GPT-4o-mini drops sharply on certain tasks (e.g., MemoryAgent HR@N=0), but the paper does not analyze possible reasons (such as prompt engineering flaws or inference hyperparameters).

7.On the Yelp dataset, agent performance is generally low (Table 2). The paper does not explore how domain-specific characteristics (such as the uniqueness of local business recommendation) might influence these results.

**Strengths Contributions:**

1.First comprehensive benchmark for LLM agent-based recommenders, addressing a critical gap in a rapidly evolving field. Prior works (e.g., Agent4Rec [11], RecMind [23]) lack standardized evaluation across diverse scenarios (cold-start, evolving interests) and multi-domain datasets.

2.Dynamic textual simulator with U-R-I network (§3.1) and scenario-task control (§3.2) is a methodological leap beyond static datasets (e.g., MovieLens). Enables interactive, adaptive agent evaluation—unachievable with traditional fixed train-test splits.

3.Modular framework (§2, Fig. 4) decouples planning/reasoning/memory components, enabling rapid prototyping (e.g., extending BaseAgent → CoTMemAgent). Lowers barriers for community adoption.

4.Real-world validation via AgentSociety Challenge (295 teams, 1,400+ submissions [9]), proving utility: 20.3% performance gain in Development Phase (§1).

5.Sustainable ecosystem: Public HuggingFace dataset + leaderboard ensures long-term usability (§1, Abstract).

6.Multi-dimensional evaluation: Tests 10+ methods across 3 paradigms (traditional, DL, agentic), 3 datasets (Amazon/Goodreads/Yelp), and 3 scenarios (classic/cold-start/evolving) (§4–5).

---

> ### Author Rebuttal · Authors · 2025-07-31
>
> **Q1: Elaborate on the challenges in constructing the benchmark and to highlight how this work complements limitations of existing evaluation frameworks.**
>
> **Response:**
>
> **Challenges in constructing the benchmark:**
>
> - **Building Interactive Environment:** We built an interactive environment by structuring public datasets into a unified schema and developing standardized information retrieval tools for agents to interact and gather personalized data.
> - **Diverse Scenarios:** A challenge is dynamically filtering data for complex scenario demands. We propose a dynamic data visibility control method, enabling flexible recommendation scenario construction.
>
> **Difference from existing evaluation frameworks:**
> Existing recommendation benchmarks ([2]-[5]) are **static and non-interactive**, making them inadequate for Recommendation Agents requiring autonomous interaction and dynamic information gathering. Our benchmark addresses this by providing an **interactive environment**. Our information retrieval tools allow agents to navigate user, review, and item networks, dynamically fetching contextual semantic information for personalized recommendations. Furthermore, we've enriched our benchmark with diverse scenarios, including classic, long/short-term, and cold-start recommendations, enabled by our retrieval tools' dynamic data and filtering capabilities (Section 3.2). This results in a more comprehensive and realistic evaluation of recommendation agent performance, mirroring real-world task scenarios.
>
> We will integrate this discussion into the Introduction.
>
> [1] Agentsociety challenge: Designing llm agents for user modeling and recommendation on web platforms.
>
> [2] Are we evaluating rigorously? benchmarking recommendation for reproducible evaluation and fair comparison.
>
> [3] Openp5: An open-source platform for developing, training, and evaluating llm-based recommender systems.
>
> [4] Beyond Utility: Evaluating LLM as Recommender.
>
> [5] https://www.recsyschallenge.com
>
>
>
> **Q2: A differential analysis is needed to clarify distinctions with similar attempts.**
>
> **Response:**
> Here's a comparative analysis of some previous attempts to clarify our unique contributions:
>
> - **Agent4Rec [1]:** This work primarily assesses generative agents for user behavior simulation. Its focus is on the fidelity of an agent's role-playing (e.g., rating, commenting) against real user behavior, not on the agent's capacity for autonomous information gathering to make recommendations.
> - **AgentCF [2]:** This proposes a multi-agent simulation approach for collaborative filtering, modeling users and items as agents. It emphasizes personalized behavior modeling within the simulation, but doesn't evaluate an agent's ability to execute an end-to-end recommendation task.
> - **AFL [3]:** While closer to our work in involving a recommendation agent interacting with an environment or user agent, AFL primarily emphasizes the agent's ability to learn from user feedback. Our work, in contrast, focuses on the agent's information retrieval and personalized analysis capabilities, which are more aligned with real-world recommendation scenarios.
>
> In summary, our benchmark is different from previous works in terms of evaluating an agent's core capabilities in autonomous, dynamic information retrieval and complex reasoning for personalized recommendation.
>
> [1] On generative agents in recommendation.
>
> [2] Agentcf: Collaborative learning with autonomous language agents for recommender systems.
>
> [3] Agentic feedback loop modeling improves recommendation and user simulation.
>
>
>
> **Q3: Need more description about the evolution of traditional recommendation methods and clarification about why existing evaluations are inadequate for agents.**
>
> **Response:**
>
> **Evolution of traditional recommendation methods:**
>
> Traditional recommender systems, initially rule-based, include methods like ItemKNN ([1]), an item-based collaborative filtering approach. Matrix Factorization (MF) ([2]) then introduced latent factors to capture user preferences and item characteristics. The field evolved with deep learning, enabling the capture of complex feature interactions. Examples include NCF [3], which uses neural networks for higher-order interactions; BPR ([4]) and BC-loss ([5]), which address ranking optimization and popularity bias, respectively. More recently, graph-based methods like LightGCN ([6]) and XSimGCL ([7]) have utilized graph convolutions and contrastive learning to enhance recommendations by aggregating neighborhood information.
>
>
>
> **Explanation about why existing evaluations are inadequate for agents**:
> Autonomous agents inherently require dynamic interaction with their environment, often involving multi-step reasoning and adaptive tool calling. Traditional evaluation frameworks based on **static data** (e.g., user-item matrices) or fixed LLM prompts, inherently **lack this interactive capability**.
> To further illustrate this, we experimented with **providing LLMs with traditional static user-item matrix data** for recommendation tasks on Amazon data. As shown below, performance was significantly lower compared to agents that dynamically collect information. This demonstrates that **static, non-interactive data fails to fully leverage an agent's ability** for autonomous information gathering and personalized reasoning, thus inadequate for agentic recommendation evaluation.
>
> | Method                                        | Qwen 2.5 | Deepseek V3 | GPT-4o-mini |
> | --------------------------------------------- | -------- | ----------- | ----------- |
> | LLM with static matrix data                   | 19.7     | 9.0         | 15.3        |
> | Autonomous Rec Agent (Baseline666) | 44.9     | 54.1        | 36.6        |
>
>
>
> [1] Item-based collaborative filtering recommendation algorithms.
>
> [2] Matrix factorization techniques for recommender systems.
>
> [3] Neural collaborative filtering.
>
> [4] BPR: Bayesian personalized ranking from implicit feedback.
>
> [5] Incorporating bias-aware margins into contrastive loss for collaborative filtering.
>
> [6] Lightgcn: Simplifying and powering graph convolution network for recommendation.
>
> [7] YXSimGCL: Towards extremely simple graph contrastive learning for recommendation.
>
>
>
> **Q4: Explain how “dynamic planning” is implemented in terms of module interactions.**
>
> **Response:**
> We want to clarify that our work primarily introduces a benchmark for recommender agents, not a new agent methodology. To help with agent development, we've adopted a modular agent design framework [1] that includes planning, reasoning, tool use, and memory.
>
> Here's how these modules interact: The **planning module** first decomposes a task into sub-tasks. These sub-tasks are then passed to the **reasoning module**, which prompts LLMs for solutions. If the LLM's internal knowledge is insufficient, the **tool use module** activates, selecting appropriate tools. The reasoning process also accesses the **memory module** for relevant past observations and experiences.
>
> The planning module's implementation is flexible, focusing on **task decomposition**. Researchers using our benchmark can flexibly explore various strategies, such as CoT planning [2] or reflection-based planning [3].  We'll update Lines 54-57 in the paper to include this detailed explanation.
>
> [1] AgentSquare: Automatic LLM Agent Search in Modular Design Space.
>
> [2] Chain-of-thought prompting elicits reasoning in large language models.
>
> [3] Self-refine: Iterative refinement with self-feedback.
>
>
>
> **Q5: The memory management mechanism lacks implementation details.**
>
> **Response:**
> Similar to the discussion about Q4, within our modular agent design framework, the memory module's implementation is also flexible, allowing users to design their own strategies. Generally, there are several ways to design memory storage. This could involve storing historical interaction records [1] or using summarized experiences [2]. For memory retrieval, common approaches include ranking and recalling based on text embedding similarity [3] or using LLMs to assess the importance of memory snippets for recall [4]. Such flexibility lets researchers explore different memory management strategies when using our benchmark.
>
> [1] ChatDev: Communicative Agents for Software Development.
>
> [2] Voyager: An open-ended embodied agent with large language models.
>
> [3] A-mem: Agentic memory for llm agents.
>
> [4] Generative agents: Interactive simulacra of human behavior.
>
>
>
> **Q6: Analyze possible reasons of the low performance of GPT-4o-mini.**
>
> **Response:** Upon careful examination, we identified the cause as a hint engineering issue: GPT-4o-mini failed to properly invoke its memory module during inference. This prevented the agent from retrieving crucial historical interactions for decision-making. We've corrected this flaw and re-ran the affected experiments. We will report the mean and variance from multiple runs in the revised paper.
>
>
>
> **Q7: Explore how domain-specific characteristics influence the performance on Yelp dataset.**
>
> **Response:** Thanks for the suggestion. We offer the following explanations about this issue:
>
> - **Data Sparsity:** As detailed in Section 4.1, Yelp dataset exhibits significantly fewer user-item interactions and a pronounced long-tail distribution. This high sparsity, across both users and items, hinders agents from capturing reliable preference patterns from limited historical data, leading to reduced recommendation accuracy.
> - **Domain-Specific Challenges:** Yelp's focus on local services (e.g., restaurants) introduces more context-dependent and geographically constrained user preferences compared to the relatively static items (products, books) on Amazon and Goodreads. Some crucial factors like location proximity or business hours are not considered in the current task setting, thus limiting the agent's effectiveness.

---

> > ### Comment · Reviewer_op5v · 2025-08-05
> > **AgentRecBench: Benchmarking LLM Agent-based Personalized Recommender Systems**
> >
> > Thank you for the comprehensive and well-structured responses. I appreciate the clarifications, and my questions have been satisfactorily addressed. I will retain my original score.

---

> > > ### Author Response · Authors · 2025-08-05
> > >
> > > Thanks for your valuable time and positive feedback. We are delighted that our responses have satisfactorily addressed your questions. Your suggestions for our work are greatly valued.

---

### Note · Authors · 2025-08-12

Dear Reviewers and ACs,

Thanks for your diligent work and insightful feedback throughout the rebuttal and discussion process. We are pleased to have the opportunity to provide a brief summary of our rebuttal and discussions.

We are pleased that all reviewers **gave encouraging evaluations and positive scores (5,4,4) in the first round** and acknowledged the core strengths of our work:

- **Novelty:** The work is **the first comprehensive benchmark for LLM agent-based recommenders** **across diverse scenarios**, addressing a critical gap in a rapidly evolving field. The benchmark adopts a **novel dataset construction** **approach**. (All reviewers)
- **Practical value:** The datasets are **open-source** and have been **successfully applied in real-world scenarios**, with **long-term usability**. The modular agentic framework **enables rapid prototyping and evaluation**, **lowering barriers for community adoption**. The paper reveals **key design insights** and provides **practical guidelines** for building better agentic recommender systems. (All reviewers)
- **Clarity:** The paper is **well-structured** with **clear language** and **appropriate visual content**.  (Reviewer yMHW)

We are pleased that our rebuttal has addressed some reviewer concerns and received positive feedback. Reviewer op5v stated that **our response is "comprehensive and well-structured"** and "**the** **questions have been satisfactorily addressed**". Reviewer FmG6 and Reviewer yMHW have also read our response with **no further concerns**. These all underscore the effectiveness and thoroughness of our responses.

Our responses are summarized as follows:

- **Reviewer op5v:**  We clarified the challenges of building our benchmark and its distinction from prior recommendation agent evaluations, explaining why traditional datasets are inadequate. We also detailed our modular agent framework design and analyzed some specific experimental results.
- **Reviewer yMHW:** We included results with standard deviations, added two advanced traditional recommendation baselines, and supplemented our discussion on general LLM agent benchmarks to better highlight our novelty.
- **Reviewer FmG6:** We clarified our approach to ensuring dataset quality, discussed the potential application scenarios for this benchmark, and discussed related techniques and benchmarks in personalized AI assistants.

Thanks again for your efforts and contributions toward a fair and insightful discussion.

---

### Decision · Program_Chairs · 2025-09-18

**Decision:**

Accept (spotlight)

**Comment:**

## Summary
This paper introduces AgentRecBench, the first comprehensive benchmark for evaluating LLM-powered agentic recommender systems. It includes: 1) A dynamic textual environment with multi-domain datasets (Yelp, GoodReads, Amazon) and dynamic data visibility control for three recommendation scenarios (classic, cold-start, evolving interests). 2)  A modular framework for developing agentic recommenders, emphasizing dynamic planning, reasoning, and memory management. 3) A systematic comparison of 10 traditional and agentic methods across three scenarios (classic, cold-start, evolving interests).

The paper demonstrates agentic systems’ superiority in adaptability and performance, alongside actionable design guidelines. Meanwhile, the benchmark’s validity is reinforced by community adoption (e.g., AgentSociety Challenge with 295 teams).

## Strengths
+ Addresses a critical gap by providing the first standardized evaluation framework for agentic recommenders, distinct from static benchmarks (e.g., MovieLens).
+ Open-sourced datasets, modular framework, and a public leaderboard enable reproducible research and community-driven progress.
+ Rigorous evaluation via testing diverse methods (traditional, DL, agentic) across scenarios, datasets, and LLMs (Qwen, DeepSeek, GPT-4o-mini), with empirical validation via a large-scale challenge.
+ Identifies key design principles (e.g., dynamic planning, memory integration) for agentic systems.


## Weakness
+ Early reviews highlighted insufficient details on benchmark construction challenges, limited statistical rigor (e.g., missing error bars), and sparse comparisons to recent non-agentic baselines.
+ Reviewers sought deeper explanations of module interactions (e.g., dynamic planning triggers) and memory management strategies.
+ Performance disparities (e.g., low Yelp scores) lacked exploration of dataset-specific factors (e.g., data sparsity).

## Rebuttal
+ Reviewer op5v: Concerns about benchmark differentiation and module interactions were addressed by clarifying the interactive environment’s uniqueness (vs. static prior works) and detailing task decomposition logic.
+ Reviewer yMHW: Statistical rigor was improved via 5-run experiments with standard deviations; modern baselines (BC-Loss, XSimGCL) were added, strengthening comparative analysis.
+ Reviewer FmG6: Dataset quality was justified through scenario-based filtering, dynamic visibility controls, and community validation. Application scenarios (e.g., user-side assistants, cold-start enhancement) were elaborated.

All reviewers acknowledged the thorough rebuttal, with concerns satisfactorily resolved.

## Justification for acceptance
This paper is recommended for acceptance as it addresses a critical gap by introducing the first standardized, interactive benchmark for evaluating agentic recommender systems across diverse scenarios. It combines methodological rigor—a novel simulator, modular framework, and extensive experiments—to derive actionable insights like memory management best practices. The work’s community impact is highlighted by publicly available tools, a leaderboard, and a successful open challenge, ensuring reproducibility and engagement. Furthermore, the work sets a new standard for evaluating adaptive, LLM-driven recommenders, making it a cornerstone for future research in this rapidly evolving field.